# Optimization of 3D Tolerance Design Based on Cost–Quality–Sensitivity Analysis to the Deviation Domain

**Kaili Yang** [1], **Yi Gan** [2], **Yanlong Cao** [1], **Jiangxin Yang** [1] and **Zijian Wu** [1,*]

1   Key Laboratory of Advanced Manufacturing Technology of Zhejiang Province, Zhejiang University, Hangzhou 310027, China
2   School of Mechanical Engineering, University of Shanghai for Science and Technology, Shanghai 200082, China
*   Correspondence: virginia@zju.edu.cn

**Abstract:** Under the new geometric product specification (GPS), a two-dimensional chain cannot completely guarantee quality of the product. To optimize the allocation of three-dimensional tolerances in the conceptual design stage, the geometric variations of the tolerance zone to the deviation domain will be mapped in this paper. The deviation-processing cost, deviation-quality loss cost, and deviation-sensitivity cost function relationships combined with the tolerance zone described by the small displacement torsor theory are discussed. Then, synchronous constraint of the function structure and tolerance is realized. Finally, an improved bat algorithm is used to solve the established three-dimensional tolerance mathematical model. A case study in the optimization of three-part tolerance design is used to illustrate the proposed model and algorithms. The performance and advantage of the proposed method are discussed in the end.

**Keywords:** tolerance allocation; small displacement torsor; tolerance zone; deviation domain; functional surface; form and position tolerances

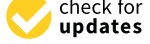



## 1. Introduction

Tolerance refers to the degree of deviation of the size, form, and position allowed in the design process. The mathematical modeling of the tolerance is to represent the tolerance in mathematical way from which the mathematical relationship of the geometric and form and position variations between the nominal size and the actual size could be obtained. The quality control in the single-part stage regarding accuracy and usability for the assembling stage will lead to a reduction of reworking and scrap. Mechanical product reliability is an important product quality factor and is dependent on different parameters among which tolerance design is an important activity. Tolerance design is usually carried out on detailed geometric entities after the structure design process [1]. With the development of computer technology, CAD/CAM plays an important role in modern industry and Computer-Aided Tolerance (CAT) design is a key technology in the integration of CAx in modern manufacturing systems. In the past decades, CAT researchers focused on the energy consumption on the customers' side, aiming to improve the quality and reduce the cost. It is also a key technology to develop automation of the industry. There are several existing works on CAT in the literature. Cao [2] provided a summary of the current research on computer-aided tolerance. Narahari Y. [3] proposed a method called DFT (design for tolerance) with a function–assembly–behavior model. Dantan J.Y. [4] put forward a methodology called ITP (Integrated Tolerance Process) to ensure tolerance traceability. He [5] studied T-maps which could automate the assignment of tolerances during design. Mantripragad [6] put forward a top-down tolerance design method that supports the assembly model. Huang [7] developed a stream-of-variation analysis (SOVA) model for three-dimensional (3D) rigid-body assemblies in a single station.

Overall, the tolerance design is mainly a multi-objective optimization problem. The three-dimensional tolerance mathematical model is the basis of tolerance design and analysis verification and plays a vital role in tolerance analysis and allocation. In recent years, there has been a lot of research on the mathematical modeling and representation of tolerances, and the mathematical models are becoming more and more adequate. Du [8] proposed a spherical multi-output Gaussian process (S-MOGP) method to model and monitor 3D surfaces to describe the geometrical deviations. Ghaderi [9] developed a Bayesian model for tolerance-reliability analysis to balance minimized cost and maximized product performance. Balamurugan [10] suggested a model considering both tolerance cost and the cost of quality loss. Feng [11] proposed an integrated parameter and tolerance design (IPTD) method based on a multivariate Gaussian process (MGP) model. Goetz [12] described a holistic methodology that supports the designer in developing a robust product layout including an initial, validated tolerance specification based on the functional requirements. Ahmad [13] studied the integrated supplier selection and tolerance allocation problems in a two-echelon make-to-order supply chain.

However, there are few studies on the correlation between the existing models and the invariance class in the new GPS standard [14]. In the new GPS standard, according to the intrinsic characteristics and positional characteristics of the geometric shape, the geometric shapes of the part is divided into seven basic invariance classes, one of which has an invariance degree. The sum of the invariance degree and the degree of variance in each invariance class is six, which provides new method for the modeling and representation of geometry shapes.

This paper is organized as follows: in Section 2, the modeling steps of the three-dimensional tolerance domain under the new GPS standard is studied and the theory of small displacement torsor (SDT) theory and homogeneous transformation matrix is introduced, which establishes the constraints of the tolerance zone represented by the torsor based on the knowledge of invariance degree and variance degree in the new GPS standard. In Section 3, the mapping of the tolerance–cost function is studied in relation to the deviation–cost function on the basis of the existing classic tolerance–cost function model and constraint conditions. The deviation parameters in the six directions of the SDT are used to represent the geometric variations of the 3D tolerance to the total cost of parts processing, instead of the tolerance–cost model represented by a single parameter. A mathematical model of processing cost, quality loss cost and sensitivity cost is established based on three-dimensional deviations. In Section 4, the concept of a functional surface is put forward, thus the conceptual structure and tolerance are combined together to establish a assembly network, form the tolerance loop, and the steps of the proposed improved bat algorithm are given to solve the deviation domain mathematical model. In Section 5 the effectiveness of the proposed 3D tolerance modeling and algorithm through a simple assembly is verified.

## 2. Parametric Modeling and Representation of 3D Tolerance Zone Based on New GPS

*2.1. Small Displacement Torsor Theory and Homogeneous Transformation Matrix (HTM) Principle*

In the 1980s, the small displacement torsor (SDT) was first proposed by Bourdet et al. [15] in metrology, and was later applied to the tolerance field by Bourdet et al. [16] in 1996. It used a point set to represent an ideal surface, replaced the size of the geometric variation tolerance zone of the surface with the spatial variation of points, and used a matrix to calculate the spatial variation of a point. To apply SDT to the field of tolerance, two basic assumptions must be satisfied: first, the rigid body assumption, the functional surface deformation of the part relative to the tolerance value is negligible; second, the small amount assumption, the nominal size of the part is larger than the tolerance value, that is, the tolerance value is a very small amount.

The small displacement torsor refers to the six independent small variations produced by the object in the space movement, so the SDT parameter represents the small deviation in

the ideal feature state and can be used to describe the position variation of a certain feature relative to its nominal feature. Therefore, the SDT can be used to model the dimensional, form, and position tolerances. The SDT consists of three translation components along the coordinate axis and three rotation components around the coordinate axis. These components can also be called torsor parameters or SDT parameters. The equation is as follows:

$$\boldsymbol{\tau} = \left\{ \begin{matrix} \rho \\ \varepsilon \end{matrix} \right\} \tag{1}$$

where $\boldsymbol{\rho} = [\alpha, \beta, \gamma]^T$ is the rotation vector and $\boldsymbol{\varepsilon} = [u, v, w]^T$ is the translation vector.

In the field of mathematics, using homogeneous coordinates has reduced the amount of matrix calculations to a large extent, and provided the conversion of a set of digital points in two-dimensional, three-dimensional, and even higher-dimensional spaces from one coordinate system to another. The homogeneous coordinate method is a method of representing n-dimensional coordinates with (n + 1)-dimensional vectors.

If a new coordinate system is obtained by first rotating the original coordinate system around the x-axis by angle $\theta_x$, then rotating around the y-axis by angle $\theta_y$, and then rotating around the z-axis by angle $\theta_z$, the new coordinate system is a homogeneous rotation transformation relative to the original coordinate system The matrix is:

$$[R] = [R_x][R_y][R_z] = \begin{bmatrix} c\theta_y c\theta_z & -c\theta_y s\theta_z & s\theta_y & 0 \\ c\theta_y s\theta_z + s\theta_x s\theta_y c\theta_z & c\theta_x c\theta_z - s\theta_x s\theta_y s\theta_z & -s\theta_x c\theta_y & 0 \\ s\theta_y s\theta_z - c\theta_x c\theta_z s\theta_y & s\theta_x c\theta_z + c\theta_x s\theta_z s\theta_y & c\theta_x c\theta_y & 0 \\ 0 & 0 & 0 & 1 \end{bmatrix}, \tag{2}$$

where $s = \sin$, $c = \cos$, $s\theta_x = \sin(\theta_x)$, $c\theta_y = \cos(\theta_y)$, and so on.

As shown in Figure 1, the coordinate system $O_{A_{xyz}}$ is transformed to $O_{B_{xyz}}$ by translation, and then transformed to $O_{C_{xyz}}$ by rotation at the origin of $O_{BC}$; the comprehensive transformation matrix is:

$$[T] = [H][R] = \begin{bmatrix} c\theta_y c\theta_z & -c\theta_y s\theta_z & s\theta_y & \Delta x \\ c\theta_y s\theta_z + s\theta_x s\theta_y c\theta_z & c\theta_x c\theta_z - s\theta_x s\theta_y s\theta_z & -s\theta_x c\theta_y & \Delta y \\ s\theta_y s\theta_z - c\theta_x c\theta_z s\theta_y & s\theta_x c\theta_z + c\theta_x s\theta_z s\theta_y & c\theta_x c\theta_y & \Delta z \\ 0 & 0 & 0 & 1 \end{bmatrix} \tag{3}$$

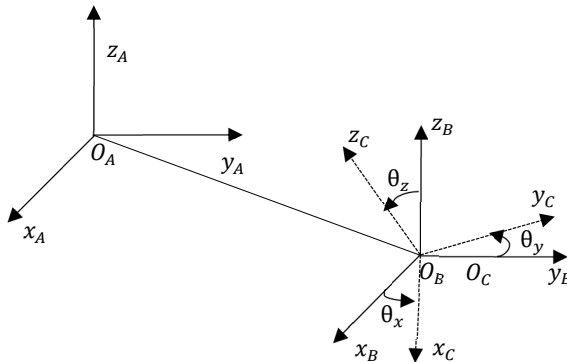

**Figure 1.** Comprehensive transformation of two different coordinate systems.

When the geometric variable $\Delta\theta$ is a small number, combined with the simplified calculation of the sine and cosine function at the small value: $\sin(\Delta\theta) = \Delta\theta$, $\cos(\Delta\theta) = 1$, then the spatial comprehensive homogeneous transformation matrix is:

$$[T] = \begin{bmatrix} 1 & -\Delta\theta_z & \Delta\theta_y & \Delta x \\ \Delta\theta_z & 1 & -\Delta\theta_x & \Delta y \\ -\Delta\theta_y & \Delta\theta_x & 1 & \Delta z \\ 0 & 0 & 0 & 1 \end{bmatrix}, \tag{4}$$

*2.2. 3D Tolerance Modeling*

2.2.1. Create a Coordinate System

The coordinate system is the prerequisite for the establishment of mathematical models, and it is also an important reference basis for the establishment of three-dimensional tolerance mathematical models. To carry out parametric modeling of the tolerance domain, firstly, the reference coordinate system needs to be determined according to the tolerance coordinate system. Secondly, the coordinate system is layered according to the level of the assembly, which mainly includes the assembly layer, part layer, and feature surface layer. A global coordinate system is built in the assembly layer, a local coordinate system is built in the part layer, and a feature coordinate system is built on the feature surface, which is mainly built on the minimum geometric datum elements (MGDE).

Therefore, for parts or surfaces with nominal symmetry of geometric features, the coordinate system is established at their symmetrical positions; the coordinate system can be established at the center point for geometric elements, and the origin of the coordinate system based on the reference point, line, and surface of MGDE generally takes itself as the center point. The establishment of the coordinate axis is based on the direction of the position change of its degree of freedom, which is perpendicular to the coordinate plane.

The establishment of a coordinate system for three-dimensional tolerances can provide an important basis for the study of the range and direction of variations in geometric elements, and also provides a basis for the tolerance accumulation calculation of the three-dimensional assembly dimension chain loop [17], which is the sequence of dimensions that form a closed circuit.

2.2.2. Kinematic Model of Three-Dimensional Tolerance Accumulation Based on HTM

Combining Equation (1), each torsor parameter of SDT can be written into the homogeneous transformation matrix (HTM), and the accumulation of three-dimensional tolerance can be realized through HTM. The expression is:

$$[T] = \begin{bmatrix} 1 & -\gamma & \beta & u \\ \gamma & 1 & -\alpha & v \\ -\beta & \alpha & 1 & w \\ 0 & 0 & 0 & 1 \end{bmatrix}, \tag{5}$$

The accumulation of tolerance in a dimensional chain, mathematically, refers to the summation of variable torsor parameters of each feature surface according to the local coordinate system. In a certain two-dimensional chain, a tolerance closed loop [18] could be described as follows:

$$\vec{T_0} = \sum_{i=1}^{n} \vec{T_i} \tag{6}$$

In a certain three-dimensional tolerance closed loop, homogeneous coordinates are used to realize the transformation of all feature surface center coordinates $(X_i, Y_i, Z_i)$ to the given tolerance demand point. The local coordinate system is converted to the global coordinate system and the components are accumulated. Considering the rotation of the feature surface's torsor parameter, the rotation accumulation is actually the summation of the rotation parameters which could be described as follows according to the rotation vector $\rho = [\alpha, \beta, \gamma]^T$

$$\begin{cases} \alpha_{all} = \sum_{i=1}^{n} \alpha_i \\ \beta_{all} = \sum_{i=1}^{n} \beta_i \\ \gamma_{all} = \sum_{i=1}^{n} \gamma_i \end{cases} \tag{7}$$

where $\alpha_{all}$, $\beta_{all}$, and $\gamma_{all}$ are the cumulation components of the rotation torsor parameters around the x, y, and z axes, respectively; $\alpha_i$, $\beta_i$, and $\gamma_i$ are, respectively, the cumulation

components of the rotation torsor parameters around the x, y, and z axes of the i-th functional surface in the closed loop of the assembly dimension chain.

For the accumulation of the translation torsor parameter, it is necessary to consider the influence of the rotation torsor parameter on it, and thus the translation accumulation could be described as follows according to the translation vector $\varepsilon = [u, v, w]^T$.

$$
\begin{cases}
u_{all} = \sum_{i=1}^{n} u_i - \sum_{i=1}^{n} \gamma_i Y_i + \sum_{i=1}^{n} \beta_i Z_i \\
v_{all} = \sum_{i=1}^{n} v_i + \sum_{i=1}^{n} \gamma_i X_i - \sum_{i=1}^{n} \alpha_i Z_i \\
w_{all} = \sum_{i=1}^{n} w_i - \sum_{i=1}^{n} \beta_i X_i + \sum_{i=1}^{n} \alpha_i Y_i
\end{cases}
\tag{8}
$$

where $u_{all}$, $v_{all}$, and $w_{all}$ are the cumulation components of the translation torsor parameters around the x, y, and z axes, respectively; $u_i$, $v_i$, and $w_i$ are, respectively, the cumulation components of the translation torsor parameters around the x, y, and z axes of the i-th functional surface in the closed loop of the assembly dimension chain.

The accumulation formula of three-dimensional tolerance torsor parameter based on HTM is as follows:

$$
[T_0] = \begin{bmatrix}
1 & -\gamma_{all} & \beta_{all} & u_{all} \\
\gamma_{all} & 1 & -\alpha_{all} & v_{all} \\
-\beta_{all} & \alpha_{all} & 1 & w_{all} \\
0 & 0 & 0 & 1
\end{bmatrix} =
$$

$$
\begin{bmatrix}
1 & -\sum_{i=1}^{n} \gamma_i & \sum_{i=1}^{n} \beta_i & \sum_{i=1}^{n} u_i - \sum_{i=1}^{n} \gamma_i Y_i + \sum_{i=1}^{n} \beta_i Z_i \\
\sum_{i=1}^{n} \gamma_i & 1 & -\sum_{i=1}^{n} \alpha_i & \sum_{i=1}^{n} v_i + \sum_{i=1}^{n} \gamma_i X_i - \sum_{i=1}^{n} \alpha_i Z_i \\
-\sum_{i=1}^{n} \beta_i & \sum_{i=1}^{n} \alpha_i & 1 & \sum_{i=1}^{n} w_i - \sum_{i=1}^{n} \beta_i X_i + \sum_{i=1}^{n} \alpha_i Y_i \\
0 & 0 & 0 & 1
\end{bmatrix},
\tag{9}
$$

### 2.2.3. The Basic Properties of the Tolerance Zone and Its Torsor Interval Description

The function, processing economy, and assembly accuracy of parts are affected by tolerances to a certain extent. The most basic element for mathematical modeling of 3D tolerances is the establishment of tolerance domains. The tolerance zone represents the mathematical meaning of 3D tolerances, including two points: (1) tolerance zone refers to the allowable range of the geometric element size, form, and position of the part. The specific common tolerance zones include the distance between two parallel planes, the area within the cylinder/cylindrical surface, the area between two radii of two concentric circles, the area between two radii of two concentric cylindrical surfaces, the distance between two parallel straight lines, the area within the sphere, etc. (2) The size, form, and position of the tolerance zone may deviate from the ideal state as the tolerance zone changes. Therefore, it is necessary to consider factors such as the size of the form, position tolerance zone, and the position boundary when modeling.

In the two-dimensional chain, dimensions are restricted in an inequality.

$$
d_L \leq d_0 \leq d_U
\tag{10}
$$

where $d_0$ is the nominal dimension, $d_L$ is the lower limit of dimension, and $d_U$ is the upper limit of dimension. $T_0 = d_U - d_L$.

As different kinds of tolerance zone are defined in the new GPS, the invariance degree and the variance degree are all given. Considering the constraints which are given by the new GPS, the tolerance zone could be represented by a homogeneous transformation matrix and the variation is restricted in the constraint inequalities based on the theory of the small displacement torsor which is shown in Table 1.

**Table 1.** Correspondence between tolerance zone and SDT model.

| Tolerance Zone | Describe the Area | Homogeneous Transformation Matrix | Constraint Inequality |
|---|---|---|---|
| Between two parallel lines |  | $T = \begin{bmatrix} 1 & -\Delta\gamma & 0 & 0 \\ \Delta\gamma & 1 & 0 & \Delta v \\ 0 & 0 & 1 & 0 \\ 0 & 0 & 0 & 1 \end{bmatrix}$ | $-\frac{t}{L} \le \Delta\gamma \le \frac{t}{L}$ <br> $-\frac{t}{2} \le \Delta v \le \frac{t}{2}$ <br> $\lvert\Delta v\rvert + \left\lvert\frac{t\Delta\gamma}{2L}\right\rvert \le \frac{t}{2}$ |
| Between two parallel curves |  | $T = \begin{bmatrix} 1 & -\Delta\gamma & 0 & \Delta u \\ \Delta\gamma & 1 & 0 & \Delta v \\ 0 & 0 & 1 & 0 \\ 0 & 0 & 0 & 1 \end{bmatrix}$ | $-\frac{t}{l_{xy}} \le \Delta\gamma \le \frac{t}{l_{xy}}$ <br> $-\frac{t}{2}n_x \le \Delta u \le \frac{t}{2}n_x$ <br> $-\frac{t}{2}n_y \le \Delta v \le \frac{t}{2}n_y$ |
| Between two parallel planes |  | $T = \begin{bmatrix} 1 & 0 & \Delta\beta & 0 \\ 0 & 1 & -\Delta\alpha & 0 \\ -\Delta\beta & \Delta\alpha & 1 & \Delta w \\ 0 & 0 & 0 & 1 \end{bmatrix}$ | $-\frac{t}{L_1} \le \Delta\alpha \le \frac{t}{L_1}$ <br> $-\frac{t}{L_2} \le \Delta\beta \le \frac{t}{L_2}$ <br> $-\frac{t}{2} \le \Delta w \le \frac{t}{2}$ <br> $\lvert\Delta w\rvert + \left\lvert\frac{L_2\Delta\alpha}{2}\right\rvert + \left\lvert\frac{L_1\Delta\beta}{2}\right\rvert \le \frac{t}{2}$ |
| Between two parallel surfaces |  | $T = \begin{bmatrix} 1 & -\Delta\gamma & \Delta\beta & \Delta u \\ \Delta\gamma & 1 & -\Delta\alpha & \Delta v \\ -\Delta\beta & \Delta\alpha & 1 & \Delta w \\ 0 & 0 & 0 & 1 \end{bmatrix}$ | $-\frac{t}{l_{yz}} \le \Delta\alpha \le \frac{t}{l_{yz}}$ <br> $-\frac{t}{l_{xz}} \le \Delta\beta \le \frac{t}{l_{xz}}$ <br> $-\frac{t}{l_{xy}} \le \Delta\gamma \le \frac{t}{l_{xy}}$ <br> $-\frac{t}{2}n_x \le \Delta u \le \frac{t}{2}n_x$ <br> $-\frac{t}{2}n_y \le \Delta v \le \frac{t}{2}n_y$ <br> $-\frac{t}{2}n_z \le \Delta w \le \frac{t}{2}n_z$ |
| Ring |  | $T = \begin{bmatrix} 1 & 0 & 0 & \Delta u \\ 0 & 1 & 0 & \Delta v \\ 0 & 0 & 1 & 0 \\ 0 & 0 & 0 & 1 \end{bmatrix}$ | $-\frac{t}{2} \le \Delta u \le \frac{t}{2}$ <br> $-\frac{t}{2} \le \Delta v \le \frac{t}{2}$ <br> $\Delta u^2 + \Delta v^2 \le \frac{t^2}{4}$ |
| Circle |  | $T = \begin{bmatrix} 1 & 0 & 0 & \Delta u \\ 0 & 1 & 0 & \Delta v \\ 0 & 0 & 1 & 0 \\ 0 & 0 & 0 & 1 \end{bmatrix}$ | $-\frac{t}{2} \le \Delta u \le \frac{t}{2}$ <br> $-\frac{t}{2} \le \Delta v \le \frac{t}{2}$ <br> $\Delta u^2 + \Delta v^2 \le \frac{t^2}{4}$ |
| Sphere |  | $T = \begin{bmatrix} 1 & 0 & 0 & \Delta u \\ 0 & 1 & 0 & \Delta v \\ 0 & 0 & 1 & \Delta w \\ 0 & 0 & 0 & 1 \end{bmatrix}$ | $-\frac{t}{2} \le \Delta u \le \frac{t}{2}$ <br> $-\frac{t}{2} \le \Delta v \le \frac{t}{2}$ <br> $-\frac{t}{2} \le \Delta w \le \frac{t}{2}$ <br> $\Delta u^2 + \Delta v^2 + \Delta w^2 \le \frac{t^2}{4}$ |

**Table 1.** *Cont.*

| Tolerance Zone | Describe the Area | Homogeneous Transformation Matrix | Constraint Inequality |
|---|---|---|---|
| Cylinder |  | $T = \begin{bmatrix} 1 & 0 & \Delta\beta & \Delta u \\ 0 & 1 & -\Delta\alpha & \Delta v \\ -\Delta\beta & \Delta\alpha & 1 & 0 \\ 0 & 0 & 0 & 1 \end{bmatrix}$ | $-\frac{t}{L} \le \Delta\alpha \le \frac{t}{L}$ <br> $-\frac{t}{L} \le \Delta\beta \le \frac{t}{L}$ <br> $-\frac{t}{2} \le \Delta u \le \frac{t}{2}$ <br> $-\frac{t}{2} \le \Delta v \le \frac{t}{2}$ <br> $\left(u + \frac{L\Delta\alpha}{2}\right)^2 + \left(v + \frac{L\Delta\beta}{2}\right)^2 \le \frac{t^2}{4}$ |
| Between two concentric cylindrical surfaces |  | $T = \begin{bmatrix} 1 & 0 & \Delta\beta & \Delta u \\ 0 & 1 & -\Delta\alpha & \Delta v \\ -\Delta\beta & \Delta\alpha & 1 & 0 \\ 0 & 0 & 0 & 1 \end{bmatrix}$ | $-\frac{t}{L} \le \Delta\alpha \le \frac{t}{L}$ <br> $-\frac{t}{L} \le \Delta\beta \le \frac{t}{L}$ <br> $-\frac{t}{2} \le \Delta u \le \frac{t}{2}$ <br> $-\frac{t}{2} \le \Delta v \le \frac{t}{2}$ <br> $\left(u + \frac{L\Delta\alpha}{2}\right)^2 + \left(v + \frac{L\Delta\beta}{2}\right)^2 \le \frac{t^2}{4}$ |

## 3. Mathematical Modeling of 3D Tolerance Allocation

### 3.1. Establishment of 3D Tolerance Mathematical Model

Based on the research of 3D tolerance zone modeling and representation, the six components of the torsor matrix are used to represent the deviation parameters in six directions, and the problem of tolerance allocation is transformed into a problem of deviation synthesis. The objective function and constraint conditions required for tolerance allocation can be represented by SDT deviation parameters. On the other hand, the tolerance zone of each functional feature surface and the assembly clearance domain of different part feature functional surfaces can also be represented by SDT deviation, as shown in Figure 2. The steps of 3D tolerance modeling are as follows:

1. By studying the SDT torsor constraint relationship of different functional feature surfaces, the tolerance zone and clearance domain between parts are mapped to the deviation domain, and the geometric variation is represented in six directions.

2. Based on the mathematical model of cost-tolerance in the classic two-dimensional dimensional chain, the single independent variable is replaced with six deviation components through torsor parameters of the deviation domain. Because the deviations in the six directions are orthogonal and linearly independent, the deviation cost can be represented by the product of each direction.

3. Next, we establish the relationship between the deviation domain and the cost function. According to Table 1, it can be found that each torsor has a constraint boundary. After combining the objective function and constraint conditions, a three-dimensional tolerance mathematical model is established.

4. In order to verify the validity and correctness of the mathematical model, a three-dimensional function graph is made under the given correlation coefficient to judge whether the model is in line with actual production and use.

5. In addition to the geometric variation constraints of the SDT torsor, it is also necessary to consider the functional requirements constraints and assembly tolerance constraints, and they should be expressed in the deviation domain.

6. Combining the objective function and constraint conditions based on the deviation domain, the three-dimensional tolerance mathematical model is built.

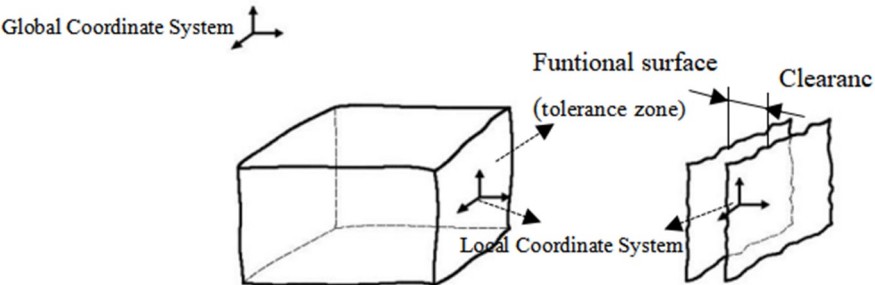

**Figure 2.** SDT assembly clearance domain.

*3.2. Processing Cost–Tolerance Model Based on HTM-SDT*

For the results of the existing research, it has been shown that the processing cost of the product is the most important part in the total cost. The total cost of the product mainly includes manufacturing and processing costs, assembly and inspection costs, and other costs in the entire life cycle. The mathematical relationship between tolerance and cost is complicated. Due to the different geometric shapes of parts, different materials, and different processing techniques, it is necessary to classify different features or specify their ranges for certain conditions. In this Section, a tolerance–cost model is discussed and presented, and its curve is drawn combining by the fitting function method in mathematics to fit a large amount of research data. Table 2 summarizes the currently proposed classical tolerance–cost models [19].

**Table 2.** Classical tolerance–cost models.

| Tolerance Types | Function Expression |
| --- | --- |
| Reciprocal model | $C(t) = a_0 + a_1 t^{-1}$ |
| Exponential model | $C(t) = a_0 + a_1 e^{-a_2 t}$ |
| Power-exponential model | $C(t) = a_0 + a_1 t^{-a_2}$ |
| Negative square model | $C(t) = a_0 + a_1 t^{-2}$ |
| Composite power-exponential and exponential model | $C(t) = a_0 + a_1 e^{-a_2 t} + a_3 t^{-a_4}$ |
| Composite linear and exponential model | $C(t) = a_0 + a_1 t + a_2 e^{-a_3 t}$ |
| Cubic model | $C(t) = a_0 + a_1 t + a_2 t^2 + a_3 t^3$ |
| Polynomial model | $C(t) = \sum_{i=0}^{n} (c_i t^i)$ |

Label: $C(t)$ is the processing cost, $t$ is the tolerance of the independent variable, and $c_i$ the constant coefficient.

It can be found that the independent variable of the classic tolerance–cost function is the tolerance $t$, and in most models, $t$ represents dimensional tolerances. However, the surface error caused by the actual processing and after the processing is more than one dimensional variation and also includes the variations of the surface shape and position of the part. The shape and position of the part are also particularly important to the processing cost of the part. Considering that it is difficult to express the geometric tolerance value with a variable, the torsor parameters in six directions of SDT are used to map the tolerance zone to the deviation domain to establish the relationship between cost and three-dimensional tolerance instead of a single independent variable $t$ [20,21].

The tolerance zone that needs to be defined on the feature surface is mapped to the deviation domain. In the tolerance modeling based on the analysis of the degree of variance of motion, each functional feature has a corresponding torsor, which is expressed as:

$$\Delta \tau = \left\{ \begin{array}{c} \rho \\ \varepsilon \end{array} \right\} = [\Delta\alpha, \Delta\beta, \Delta\gamma, \Delta u, \Delta v, \Delta w]^{\mathrm{T}}, \tag{11}$$

Therefore, we could transfer the cost–tolerance function to a function of cost and deviations as we represent tolerance as deviation in SDT:

$$C(t) \rightarrow C\{\Delta\alpha, \Delta\beta, \Delta\gamma, \Delta u, \Delta v, \Delta w\}, \tag{12}$$

The cost–tolerance function with only one independent variable is expressed as a cost–deviation function with six torsor variables. These torsor parameters are related to the functional feature surface and are mutually orthogonal and linearly independent relative to the deviation coordinate system in the deviation space. Therefore, the six torsor parameters of the processing cost can be expressed as a separate product, and Equation (12) can be further written as:

$$C(t) = C_1(\Delta\alpha) \cdot C_2(\Delta\beta) \cdot C_3(\Delta\gamma) \cdot C_4(\Delta u) \cdot C_5(\Delta v) \cdot C_6(\Delta w), \tag{13}$$

where $C_1(\Delta\alpha)$, $C_2(\Delta\beta)$, and $C_3(\Delta\gamma)$ are the processing costs required to overcome the rotation error and $C_4(\Delta u)$, $C_5(\Delta v)$, and $C_6(\Delta w)$ are the processing cost required to overcome translational errors. In addition, if the feature is constant in a certain direction, it will not affect the cost. In this case, the cost is regarded as 1. For example, the rotation parameter direction of a ball in the coordinate system where the origin is the center of the sphere is constant, that is, $\Delta\alpha = \Delta\beta = (\Delta\gamma) = 0$, then $C_1(\Delta\alpha) = C_2(\Delta\beta) = C_3(\Delta\gamma) = 1$.

In order to verify that the model conforms to the variation relationship between processing costs and tolerances, a plane is used to verify and illustrate as an example. To simplify the calculation process, take two end faces of a certain part to form the loop, and study the situation where only one side of the tolerance produces geometric constraints, as shown in Figure 3.

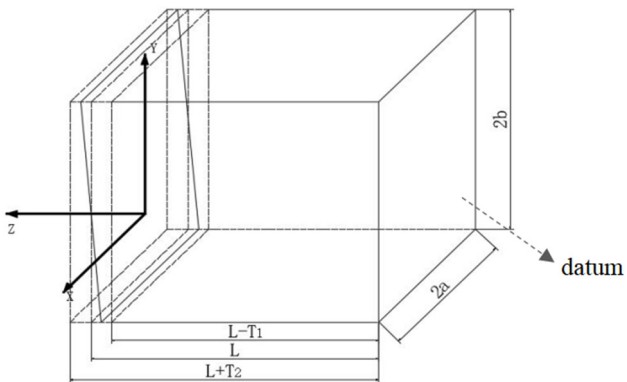

**Figure 3.** Dimension tolerance zone of a plane.

The plane's translation along the x and y axes and rotation around the z axis are constant, so it has three invariance degrees and three degrees of freedom. The torsor and processing cost function model are expressed as follows:

$$\Delta\tau = \left\{ \begin{matrix} \rho \\ \varepsilon \end{matrix} \right\} = [\Delta\alpha, \Delta\beta, 0, 0, 0, \Delta w]^T, \tag{14}$$

$$C(t) = C_1(\Delta\alpha) \cdot C_2(\Delta\beta) \cdot C_6(\Delta w), \tag{15}$$

In regard to the rotation deviation of a rectangular plane, it can be seen from Figure 3 that the deviation domains generated by rotation around the x axis and the y axis are symmetrical, and the function models of $C_1(\Delta\alpha)$ and $C_2(\Delta\beta)$ are the same, so the cost of the rotation deviation is also the same. We use $\Delta\theta$ to represent $\Delta\alpha$ and $\Delta\beta$ (see Equation (16) for details) and merge them into $C_t(\Delta\theta)$:

$$\Delta\theta = d_1 + d_2 \cdot \Delta\alpha + d_3 \cdot \Delta\beta, \tag{16}$$

$$C(t) = [C(\Delta\theta)]^2 \cdot C_6(\Delta w), \tag{17}$$

where $d_1$, $d_2$, and $d_3$ are random constants.

By omitting the quadratic term of Equation (17), it can be rewritten as:

$$C(t) = C_t(\Delta\theta) \cdot C_6(\Delta w), \tag{18}$$

where $C_t(\Delta\theta)$ is the processing cost for rotation deviations caused by controlling the plane and $C_6(\Delta w)$ is the processing cost for translation deviations caused by controlling the plane.

The reference coordinate system is established according to existing rules. The coordinate origin of the rectangular feature surface is located at the center point of the plane. We know that the maximum deviation of this plane is located at the four vertices. Therefore, the deviation change in the z axis direction is studied. If the tolerance zone is known as $[-T_1, T_2]$, the coordinates at the four vertices are expressed as $(a,b,0)$, $(a,-b,0)$, $(-a,-b,0)$, and $(-a,b,0)$, the constraint model of torsor deviation in the Z-axis direction at the four vertices can be obtained by Equations (7) and (8):

$$\begin{cases} -T_1 \leq \Delta w - a\Delta\beta + b\Delta\alpha \leq T_2 \\ -T_1 \leq \Delta w - a\Delta\beta - b\Delta\alpha \leq T_2 \\ -T_1 \leq \Delta w + a\Delta\beta - b\Delta\alpha \leq T_2 \\ -T_1 \leq \Delta w + a\Delta\beta + b\Delta\alpha \leq T_2 \end{cases}, \tag{19}$$

In order to get the variation trend of the objective function value caused by the change in the independent variable in the functional relationship, we assume that the plane is a square with a side length of 2; thus, Equation (19) can be abbreviated as:

$$\begin{cases} -T_1 \leq \Delta w - \Delta\beta + \Delta\alpha \leq T_2 \\ -T_1 \leq \Delta w - \Delta\beta - \Delta\alpha \leq T_2 \\ -T_1 \leq \Delta w + \Delta\beta - \Delta\alpha \leq T_2 \\ -T_1 \leq \Delta w + \Delta\beta + \Delta\alpha \leq T_2 \end{cases}, \tag{20}$$

The three unknown parameters of $\Delta w$, $\Delta\beta$, and $\Delta\alpha$ in the three directions of x, y, and z on the three-dimensional orthogonal coordinate axis are brought into consideration. The function graph of the above formula is shown in Figure 4.

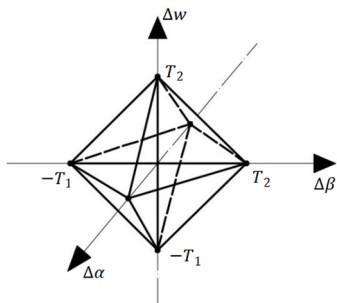

**Figure 4.** Envelope graph of parameter value of the plane tolerance zone.

The diamond-like area in the figure is the value range of the three torsor components in the deviation domain. If the three-dimensional picture is drawn, let $[-T_1, T_2] = [-0.1, 0.1]$, then Equation (20) can be written as the area surrounded by eight planes:

$$\begin{cases} \Delta w = \Delta\alpha - \Delta\beta + 0.1, \Delta w = \Delta\alpha - \Delta\beta - 0.1 \\ \Delta w = \Delta\alpha + \Delta\beta + 0.1, \Delta w = \Delta\alpha + \Delta\beta - 0.1 \\ \Delta w = -\Delta\alpha + \Delta\beta + 0.1, \Delta w = -\Delta\alpha + \Delta\beta - 0.1 \\ \Delta w = -\Delta\alpha - \Delta\beta + 0.1, \Delta w = -\Delta\alpha - \Delta\beta - 0.1 \end{cases}, \tag{21}$$

It can be seen from Figure 5 that the area enclosed by the eight planes is similar to the combination of two pyramids, so that the modeling method can map the geometric variation of the tolerance zone to the deviation space.

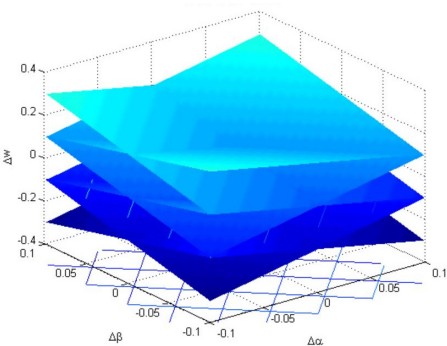

**Figure 5.** Three-dimensional drawing of eight planes.

To study the mapping of the cost–tolerance model to the cost–deviation model, it is assumed that the mathematical model of the plane takes the reciprocal model that is $C(t) = a_0 + a_1|t|^{-1}$. According to the above calculation process, the cost–deviation function for the torsor of the plane is established:

$$C_i(\Delta\tau) = a_{0\tau} + a_{1\tau}|\Delta\tau|^{-1}, \tag{22}$$

where $a_{0\tau}$ and $a_{1\tau}$ are constant coefficients.

According to Equation (18), Equation (22) can be expressed as:

$$C(t) = C_t(\Delta\theta)\cdot C_6(\Delta w) = [a_{0t} + a_{1t}/|\Delta\theta|]\cdot[a_{0\Delta w} + a_{1\Delta w}/|\Delta w|], \tag{23}$$

Equation (20) can be further transformed into:

$$\begin{cases} -T_1 \leq \Delta w + \Delta\theta \leq T_2 \\ -T_1 \leq \Delta w - \Delta\theta \leq T_2 \end{cases}, \tag{24}$$

where $\Delta\theta = \Delta\alpha + \Delta\beta$ or $\Delta\theta = \Delta\alpha - \Delta\beta$.

To draw the trend graph of the three-dimensional space function of (C—$\Delta\theta$—$\Delta w$), let $a_{0t} = a_{1t} = a_{0\Delta w} = a_{1\Delta w} = 1$ and $T_1 = T_2 = 0.1$. Combining Equations (23) and (24), we can get:

$$\begin{cases} C(\Delta\theta, \Delta w) = [1 + \frac{1}{|\Delta\theta|}]\cdot[1 + \frac{1}{|\Delta w|}] \\ -0.1 \leq \Delta\theta + \Delta w \leq 0.1 \\ -0.1 \leq \Delta\theta - \Delta w \leq 0.1 \end{cases}, \tag{25}$$

Using MATLAB to draw the graph of Equation (25) in three-dimensional space with, the correlation between the deviation and processing cost can be roughly obtained from the span, as shown in Figure 6.

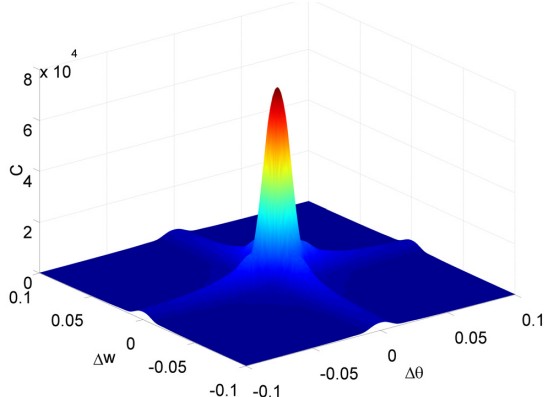

**Figure 6.** Function graph between processing cost and deviation domain.

As shown in Figures 6 and 7, the smaller the deviation is, the greater the processing cost would be, which is consistent with the monotonicity of the tolerance–cost function. The mathematical model of cost–deviation is actually an extension of the cost–tolerance model, which fully considers the geometric errors caused by the degree of variance of the part movement in six directions. Therefore, it also refers the mapping of the processing cost function of the tolerance zone to the deviation domain represented by torsor parameters.

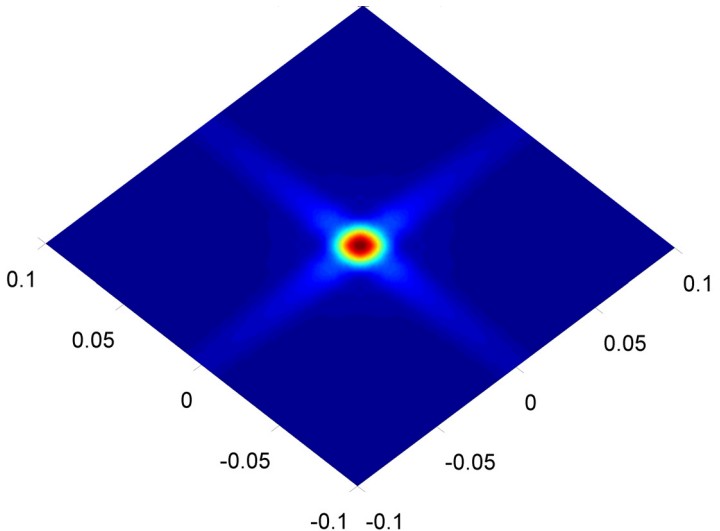

**Figure 7.** Top view of function graph between processing cost and deviation domain.

### 3.3. Quality Loss–Tolerance Model Based on HTM-SDT

When studying the impact of tolerance zone variations on cost, tolerance zone not only affects the cost of design and manufacturing, but also affects the quality of products that are designed and manufactured. In this paper, the concept of a quality loss function which was proposed by Dr. Genichi Taguchi [22] will be further discussed. Dr. Genichi Taguchi built the modeling from the perspective of the user who purchases the product. Compared with the traditional concept of quality, which is to think only from the perspective of the producer, the quality is limited by the upper and lower limit values of the tolerance, and it is qualified as long as it meets the tolerance area specified by its size. This view is correct, but the concept of quality loss improved the production demand of the product and reduced the loss rate of the product.

The absolute value of the deviation is inversely proportional to the quality and directly proportional to the loss. The expression of the quality loss function L is then defined as:

$$L(y) = k(y - m)^2, \tag{26}$$

where y is the value of quality characteristic, m is the target value, and k is the proportionality constant or cost coefficient which depends on the cost at the specification limits and the width of the specification. A graphical representation for the quality loss function for nominal-the-best case is shown in Figure 8.

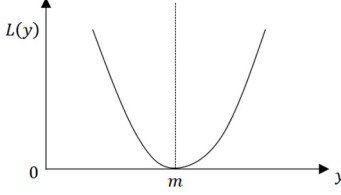

**Figure 8.** Graphical representation for quality loss function for nominal-the-best case.

In Equation (26), $(y - m)$ indicates the dimensional tolerance T during processing. Generally, there are two types of dimensional tolerances—bilateral and unilateral [23]; then:

$$L(T) = \begin{cases} k_1 \frac{T^2}{4}, & \text{bilateral tolerance} \\ k_2 T^2, & \text{unilateral tolerance} \end{cases} \tag{27}$$

where $k_1$ and $k_2$ are the coefficient of bilateral tolerance and unilateral tolerance.

The quality loss function for smaller-the-better characteristics means that the smaller the tolerance is, the better the quality would be. The function graph is similar to the right graph of the $y = m$ line in Figure 8, and the m point should be set as the origin. Its function expression is:

$$P = k \cdot \Delta t^2 \tag{28}$$

$$k = \frac{P}{\Delta p^2} \tag{29}$$

where $\Delta p$ is the deviation of product allowed by the user and P is the loss when the deviation reaches the deviation allowed by the user.

Substituting Equation (29) into Equation (27) and assuming that the tolerance is bilateral and symmetrical, then:

$$L_s(T) = \frac{PT^2}{4\Delta p^2} \tag{30}$$

For easy calculation and observation, set $\frac{PT^2}{4\Delta p^2} = b$, then:

$$L_s(T) = bT^2 \tag{31}$$

The quality loss function has only one independent variable which is the tolerance value. The influence of the variation of the surface shape and position error on the product quality is not considered, but these factors are also very important for the quality loss cost. On the basis of the research on the relationship between processing cost and deviation in the previous section, using SDT parameters to establish the mapping relationship between the cost of quality loss in six directions and the deviation domain, then:

$$L(T) \rightarrow L\{\Delta\alpha_l, \Delta\beta_l, \Delta\gamma_l, \Delta u_l, \Delta v_l, \Delta w_l\} \tag{32}$$

Following the same considerations as for Equation (13), we can get:

$$L(T) = L_1(\Delta\alpha_l) \cdot L_2(\Delta\beta_l) \cdot L_3(\Delta\gamma_l) \cdot L_4(\Delta u_l) \cdot L_5(\Delta v_l) \cdot L_6(\Delta w_l) \tag{33}$$

where $L_1(\Delta\alpha_l)$, $L_2(\Delta\beta_l)$, and $L_3(\Delta\gamma_l)$ are the quality loss costs required to overcome the rotation error and $L_4(\Delta u_l)$, $L_5(\Delta v_l)$, $L_6(\Delta w_l)$ are the quality loss costs required to overcome the translation error. Like the processing cost, if the rotation or translation of a feature surface in a certain direction is constant, the deviation–cost component value is 1.

In order to illustrate the rationality of the quality loss–deviation model, we will further verify the above-mentioned plane tolerance zone example, map the quality loss cost of the plane tolerance zone to the deviation domain, and write the quality loss function model of the plane as:

$$L(T) = L_1(\Delta\alpha_l) \cdot L_2(\Delta\beta_l) \cdot L_6(\Delta w_l) \tag{34}$$

Set $\Delta\theta_l = q_1 + q_2 \cdot \Delta\alpha_l + q_3 \cdot \Delta\beta_l$, and $q_1$, $q_2$, and $q_3$ as any constant, the above equations can be written as:

$$L(T) = L(\Delta\theta_l)^2 \cdot L_6(\Delta w_l) \tag{35}$$

Omitting the quadratic term of Equation (35), we can get:

$$L(T) = L_T(\Delta\theta_l) \cdot L_6(\Delta w_l) \tag{36}$$

where $L_T(\Delta\theta_1)$ is the quality loss cost for rotation deviation caused by controlling the plane and $L_6(\Delta w_1)$ is the quality loss cost for translation deviation caused by controlling the plane.

To map the tolerance zone of the quality loss cost to the deviation domain, and taking bilateral and symmetric tolerances into consideration, then the corresponding mathematical model of the deviation domain torsor parameters is:

$$L_i(\Delta\tau) = b_{0\tau} \cdot |\Delta\tau|^2 \tag{37}$$

where $b_{0\tau}$ is a constant coefficient.

According to Equation (31), (37) and (40) can be written as:

$$L(T) = L_T(\Delta\theta_1) \cdot L_6(\Delta w_1) = b_{0T} \cdot |\Delta\theta_1|^2 \cdot b_{1\Delta w_1} \cdot |\Delta w_1|^2 \tag{38}$$

From Equation (24), the torsor constraint inequality of the quality loss function is written as:

$$\begin{cases} -T_1 \le \Delta\theta_1 + \Delta w_1 \le T_2 \\ -T_1 \le \Delta\theta_1 - \Delta w_1 \le T_2 \end{cases} \tag{39}$$

where $\Delta\theta_1 = \Delta\alpha_1 + \Delta\beta_1$ or $\Delta\theta_1 = \Delta\alpha_1 - \Delta\beta_1$.

To draw the trend graph of the three-dimensional space function of $(L—\Delta\theta_1—\Delta w_1)$, let $b_{0T} = b_{1\Delta w_1} = 1$, $T_1 = T_2 = 0.1$. Combining Equations (38) and (39), we can get:

$$\begin{cases} L(\Delta\theta_1, \Delta w_1) = |\Delta\theta_1|^2 \cdot |\Delta w_1|^2 \\ -0.1 \le \Delta\theta_1 + \Delta w_1 \le 0.1 \\ -0.1 \le \Delta\theta_1 - \Delta w_1 \le 0.1 \end{cases} \tag{40}$$

Using MATLAB to draw the graph of Equation (40) in three-dimensional space, the relationship between deviation and quality loss cost can be roughly obtained from the span, as shown in Figure 9.

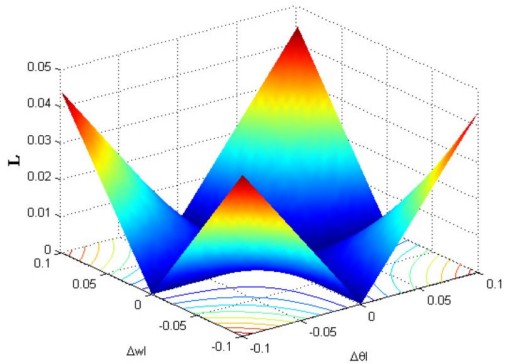

**Figure 9.** Function graph between quality loss cost and deviation domain.

According to Figures 9 and 10, the smaller the deviation is, the lower the quality loss cost is. This monotonicity is consistent with the tolerance–quality loss function. On the basis of a single independent variable, the modified model takes the six torsor parameters into account, which are the geometric errors in the directions of the six independent variables. The quality loss function of the tolerance zone is mapped to the deviation domain of the SDT parameters. The example verification shows that the model conforms to the actual design and production process.

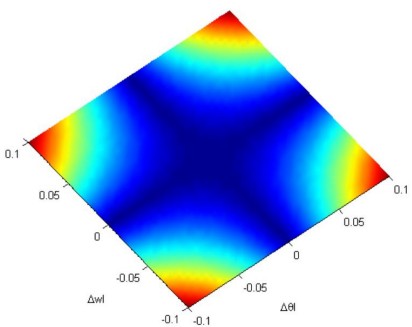

**Figure 10.** Top view of function graph between quality loss cost and deviation domain.

*3.4. Sensitivity–Tolerance Model Based on HTM-SDT*

The sensitivity–tolerance function model is based on the minimum sensitivity design method. On the basis of the processing cost–tolerance function model, the characteristics of sensitivity are used to ensure that the processing cost and quality loss cost are small while considering the influence of product resistance to designed tolerance variations. The commonly used sensitivity function is based on the processing cost–function model; using Taylor series expansion at the mean value of the tolerance t (independent variable), the mathematical expression is as follows:

$$\Delta C(t) = \frac{\partial C(t)}{\partial t_i} \cdot \Delta t + \frac{\partial C^{(2)}(t)}{2! \partial t_i} \cdot \Delta t^2 + \cdots + \frac{\partial C^{(n)}(t)}{n! \partial t_i} \cdot \Delta t^n \tag{41}$$

where $\Delta t$ is the difference between the independent variable t and the mean tolerance.

In order to facilitate the study of the law for mapping the sensitivity–tolerance zone model to the sensitivity–deviation domain model, the high-order terms of the above equation are removed, and the first-order partial derivative is taken as the first approximation, and then the sensitivity–tolerance function of product processing is obtained according to the probability method:

$$\Delta C(t) = \frac{\partial C(t)}{\partial t_i} \cdot \Delta t \tag{42}$$

Referring to the modeling process of mapping the processing cost–tolerance zone to the deviation domain, the sensitive SDT deviation parameters and the sensitivity–cost function are expressed as:

$$\Delta C(t) = \Delta C_1(\Delta \alpha) \cdot \Delta C_2(\Delta \beta) \cdot \Delta C_3(\Delta \gamma) \cdot \Delta C_4(\Delta u) \cdot \Delta C_5(\Delta v) \cdot \Delta C_6(\Delta w) \tag{43}$$

where $\Delta C_1(\Delta \alpha)$, $\Delta C_2(\Delta \beta)$, and $\Delta C_3(\Delta \gamma)$ are the costs of sensitivity required to overcome the rotation error and $\Delta C_4(\Delta u)$, $\Delta C_5(\Delta v)$, and $\Delta C_6(\Delta w)$ are the costs of sensitivity required to overcome the translation error.

Since the sensitivity is a further constraint objective function problem after the processing cost mathematical model is established, its torsor parameters are all established on the basis of the processing cost function.

To establish a mapping from the sensitivity cost of tolerance zone to the sensitivity cost of the deviation domain for the plane, and from Equation (22), the sensitivity–cost function of the plane deviation is:

$$C_i(\Delta \tau) = -c_{1\tau} |\Delta \tau|^{-2} \cdot \Delta \tau_t \tag{44}$$

where $c_{1\tau}$ is a constant coefficient.

According to Equation (23), we can get:

$$\Delta C(t) = \Delta C_t(\Delta \theta) \cdot \Delta C_6(\Delta w) = \left[ -c_{1t} |\Delta \theta|^{-2} \right] \cdot \left[ -c_{6\Delta w} |\Delta w|^{-2} \right] \tag{45}$$

To draw the trend graph of the three-dimensional space function of $(\Delta C—\Delta\theta—\Delta w)$, let $c_{1t} = c_{6\Delta w} = 1$ and $T_1 = T_2 = 0.1$. Combining Equations (44) and (24), we can get:

$$\begin{cases} \Delta C(\Delta\theta, \Delta w) = \left[\frac{1}{|\Delta\theta|^2}\right] \cdot \left[\frac{1}{|\Delta w|^2}\right] \\ -0.1 \leq \Delta\theta + \Delta w \leq 0.1 \\ -0.1 \leq \Delta\theta - \Delta w \leq 0.1 \end{cases} \tag{46}$$

Using MATLAB to draw the graph of Equation (46) in three-dimensional space, the relationship between deviation and sensitivity cost can be roughly obtained from the span, as shown in Figure 11.

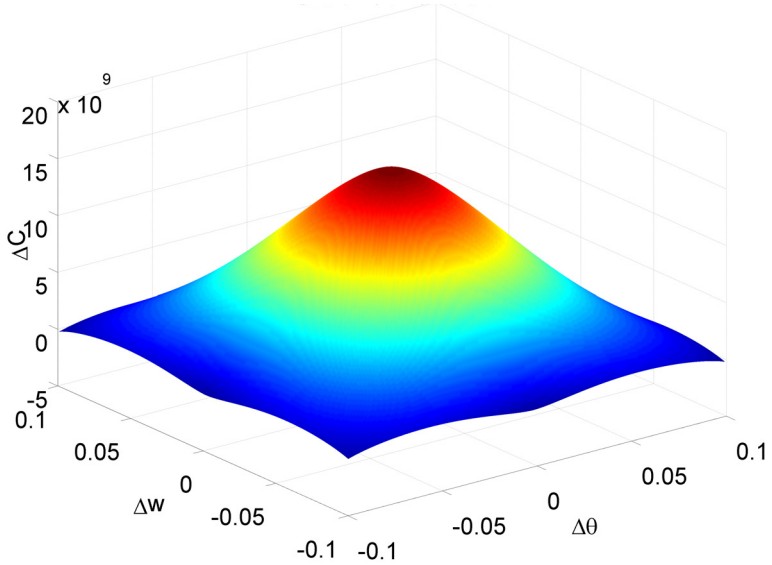

**Figure 11.** Function graph between sensitivity and deviation domain.

From Figures 11 and 12, it can be found that the trend of the function graph is the same as the processing cost–deviation domain function graph, which is also in line with the actual production law. The smaller the tolerance value is, the higher the processing cost is. The monotonicity of the sensitivity function is the same as the processing cost function. The higher the cost of resisting the variation of tolerance geometric factors is, the lower the corresponding quality loss cost is. Therefore, when considering sensitivity factors, the manufacturability of the designed product can be increased. The verification results show that the sensitivity–cost modeling in the six degrees-of-freedom direction of 3D tolerance zone meets the actual production requirements.

*3.5. Constraints*

3.5.1. Constraints of Machining Capacity and Geometric Function

The tolerance value of each component link of the assembly dimension chain cannot exceed the actual range of processing capacity; that is, to ensure the processing economy, each geometric tolerance has its own processing capacity range, and the tolerance can be mapped to the deviation domain and expressed as follows:

$$t_{imin} \leq f(\Delta\tau_i) \leq t_{imax} \tag{47}$$

where $t_{imin}$ is the minimum machining capability tolerance of the i-th component link; $\Delta\tau_i$ is the deviation value of each SDT parameter of the i-th component link; and $t_{imax}$ is the maximum machining capacity tolerance of the i-th component link.

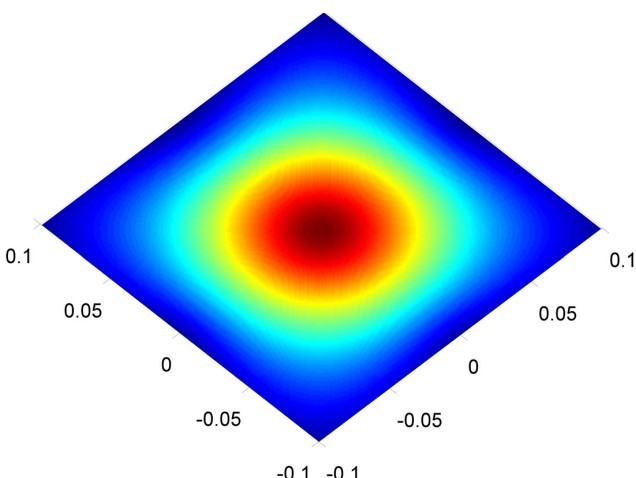

**Figure 12.** Top view of function graph between sensitivity and deviation domain.

The geometric function requirement (GFR) is a constraint given by the designer to achieve a certain performance. It represents the distance between two geometric function elements that play a practical role in an assembly part, and it can also represent the orientation of geometric function elements relative to a certain datum coordinate. The tolerance is mapped to the deviation domain, so the geometric function requirement constraint is transformed into the SDT parameters constraint inequality:

$$f(\Delta\tau) \leq \text{GFR} \tag{48}$$

3.5.2. Constraints of Assembly Tolerance

After the parts are designed and produced, to meet the functional requirements, most of them will be assembled. The essence of assembly is to constrain the parts and prevent over-constraint. Drawing on the ideas of the extreme value method and root-mean-square error (RMSE) method, which are mainly used in tolerance analyses, and thinking in reverse, the allocation of the component links should be carried out under the premise of meeting the requirements of the closed links, and the cumulative value of the allocated tolerance of each component link should be less than the function requirement. In other words, when the functional requirement is equal to the cumulative value of component links, meeting the functional requirement of part assembly means that it can be assembled.

The connection and transfer between two parts is realized on the basis of fit and contact. The "tool" used is the functional feature surface. The geometric variations of each surface or the clearance between the two surfaces can be represented by SDT parameters, and the assembly must have a closed loop. Therefore, the cumulative equation of SDT parameters can be established based on the loop and the allowable geometric variation domain which reflects that the functional feature surface of the part is established.

According to Equation (9), assuming that the assembly tolerance is **G**, the assemblability equation is defined by the assembly tolerance constraint as:

$$\{\boldsymbol{\Omega}_k, \mathbf{D}_k\}^T = \mathbf{G} \tag{49}$$

where $\boldsymbol{\Omega}_k = \{\alpha_{all}, \beta_{all}, \gamma_{all}\}^T$ and $\mathbf{D}_k = \{u_{all}, v_{all}, w_{all}\}^T$.

## 4. Steps of Optimal Allocation for 3D Tolerance Based on Modified Bat Algorithm

*4.1. Basic Concepts of Functional Structural Tolerance*

In the product conceptual design stage, optimal tolerance allocation research is carried out, and the assembly dimensional chain must be generated. Since the conceptual design stage is different from the detailed design stage, there is no actual solid model, and the generation method of the dimensional chain is different from the existing automatic generation

methods of dimensional chains. Therefore, this article mainly studies the establishment of tolerance loops through geometric elements, and the conceptual design must involve structural design, so functional surface in the structural design is considered. Based on the above reasons, this paper starts from restricting the degree of variance of geometric variations, establishes a relationship network of the part functional surface, determines the tolerance loop on the basis of the network, and proposes several rules for complex assemblies.

Functional surface is the intermediate medium between function and structure, and it is also a bridge connecting structure and tolerance. The technologically and topologically related surface (TTRS) theory proposed by Clement [24] and Salomons [25] is the embodiment of the functional surface of parts. A TTRS represents a pair of functional surfaces that are connected to each other due to their functions. The TTRS theory divides the functional surfaces of parts into seven basic types. At the same time, Clement and Salomons put forward the concept of minimum geometric datum elements (MGDE) in order to express the connection between various functional surfaces. MGDE refers to the minimum geometric elements of reference points, reference lines, and reference surfaces for positioning functional surfaces in three-dimensional space, and it is conducive to constrain the positional relationship between geometric bodies in Euclidean space. Table 3 shows the corresponding relationship between the basic functions of TTRS and MGDE [26].

**Table 3.** MGDE of functional surfaces.

| Functional Surfaces | Sphere | Plane | Cylinder | Helical | Rotational | Prismatic | General |
|---|---|---|---|---|---|---|---|
| MGDE | point | plane | line | Point and line | Point and line | Line and plane | Point and line and plane |
| Degrees of invariances | 3 rotations | 1 rotation 2 translation | 1 rotation 1 translation | 1 helical displacement | 1 rotation | 1 translation | None |
| Reference element | sphere center | plane | cylinder axis | Rotation axis and points on the surface | Rotation axis and points on the surface | Straight line in translation direction and plane in determined direction | Any combinations of surface elements |

An assembly hierarchy decomposition diagram is drawn combined with the above content as shown in Figure 13. Since structure design based on function also contains the design of dimensions and tolerances, functional surfaces and tolerances have big influence on each other. Therefore, it is necessary to use the functional surfaces to synchronize tolerance design in the early design stage. In order to realize the optimal allocation of tolerances in the process of product design, the premise is to establish a function–structure–tolerance mapping model, so that the structural design can meet the functional requirements of the product as well as ensuring the tolerance design can meet the geometric functional requirements. Based on the top-down design principle, the functional surface connects the structure and tolerance of the parts and plays an important role as shown in Figure 13.

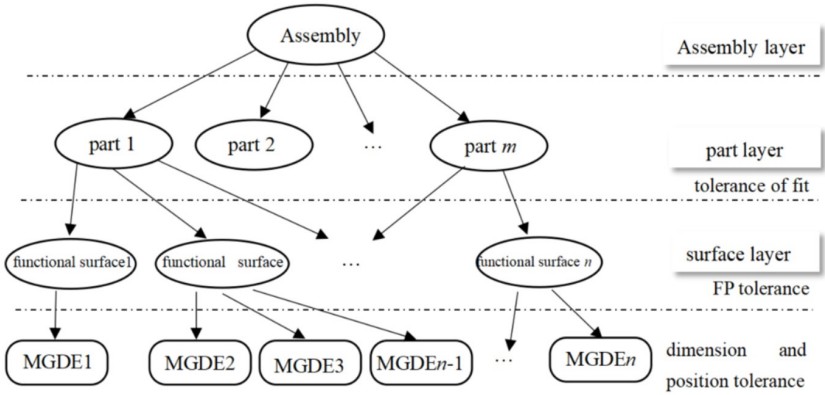

**Figure 13.** Assembly hierarchy diagram of product.

### 4.2. Construction of Assembly Tolerance Loop Based on Functional Surface

A tolerance loop is an assembly tolerance transfer chain that needs to be used in the conceptual design stage. It is a closed loop that describes the constraint relationship between nominal geometry and variational geometric constraints using diagrams and other methods on the basis of separation from the solid model. There is no uniform form of tolerance loops. Many loops can be formed in an assembly. As long as it starts from a certain local coordinate system to the next coordinate system, and the loop can be passed back to the original coordinate system, it can be called a tolerance loop. In regard to deviation propagation and accumulation in 3D space introduced in Section 2.2, it is completely applicable in the tolerance loop, and will not be repeated here.

Figure 13 clearly shows the relationship between the functional surface of each part in the assembly and the MGDE. The establishment of the tolerance loop is based on the constraint relationship of each part in the assembly, and there are three connection methods between functional feature surfaces: first, the two functional surfaces belong to different parts; second, the two functional surfaces belong to different positions of the same part; and third, the same surface of the same part, which is used in the consideration of essential deviation. In the first case, the clearance domain of geometric tolerance is generated, the connection circuit diagram between the functional surfaces of the parts in assembly is represented in Figure 14:

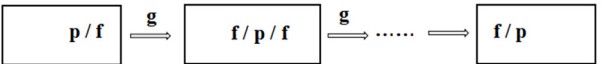

**Figure 14.** Functional surface transfer chain of parts.

In Figure 14, p refers to parts, f refers to functional surfaces, and g refers to assembly clearance. In order to facilitate the observation of the total tolerance loop of the parts so that the tolerance chain which best meets the requirements can be selected based on the diagram of the assembly hierarchy, the assembly network is established on the basis of the relationship between the part layer and the surface layer, with large circles representing parts and small circles representing functional surfaces. Arrows are used to point to the contacting or matching functional surfaces from the fixed surface to the positioning surface, and the functional requirements are connected with a dotted line. One generic case of a functional surface constraint network between three parts A, B, and C is shown in Figure 15.

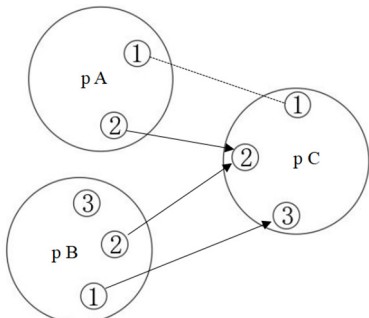

**Figure 15.** Functional surfaces of parts.

On the basis of the part functional surface network, to determine the tolerance loop, it is mainly divided into the following steps:

- Determine the geometric constraints and degrees of freedom of the two functional surfaces for functional requirements.
- Select the part where one of the functional surfaces is located, and filter out the surface of the part that participates in the constraint degree of freedom.
- Remove irrelevant part feature surfaces.

- Continue the above steps to find the next part and its surface that has geometric constraints with the functional surface of the part until it returns to the original functional surface.

Therefore, the assembly tolerance loop can be expressed as:

$$L = \{(p, f), g, [(f, p, f), g] \cdots (f, p)\} \tag{50}$$

where (p, f) refers to the deviation of the assembly function surface f on the part p.

In Figure 15, the numbers of ① ② ③ refer to the functional surfaces in different parts. In an assembly with many tolerance loops, the establishment of loops must ensure that the accumulation of errors is minimized while meeting functional requirements, and the loop cannot be under-constrained or over-constrained. The following rules are made in this article:

Rule 1: The tolerance loop must be established based on MGDE and all MGDE must be involved in at least one tolerance loop.

Rule 2: MGDE of the same functional surface can no longer be related to each other.

Rule 3: After satisfying the above rules, try to choose a tolerance circuit that involves fewer functional surfaces.

The creation method of the above tolerance loop conforms to the concept of structural design. It does not require a physical assembly drawing to generate a tolerance chain. It uses the relationship between geometric constraints and degrees of freedom of the functional surface to generate a tolerance loop on the basis of the assembly network.

### 4.3. Optimal Allocation for 3D Tolerance Based on Modified Bat Algorithm

Tolerance allocation is one of the key steps of tolerance optimal design, and it is usually carried out with the structural design of the product. The first two subsections of this section are about the conceptual structure design, and the relationship between structure and tolerances are established through the functional surface. Predecessors' research on the tolerance allocation mostly focused on two-dimensional dimensions [27], and there are many studies on reallocation in the detailed design stage. This paper proposes to optimize the allocation of three-dimensional tolerances in the conceptual design stage.

The optimal allocation of tolerances is to allocate tolerance values of each assembly dimension chain, aimed at minimizing the total cost as the objective function and the assembly success rate and manufacturing requirements on the premise of meeting product geometric functional requirements and assembly requirements. This is one of the core steps of CAT (computer-aided tolerance) calculation. The three-dimensional tolerance allocation problem can be transformed into a deviation domain optimal allocation problem in this paper.

The mathematical model of the deviation optimization is determined. First, establish the objective function relational expression based on the SDT deviation parameters:

$$\begin{cases} C(t) = \sum_i \sum_j C(\Delta\tau_{ij}) \\ C_L(t) = \sum_i \sum_j L(\Delta\tau_{ij}) \\ \Delta C(t) = \sum_i \sum_j \Delta C(\Delta\tau_{ij}) \end{cases} \tag{51}$$

where $C(t)$, $C_L(t)$, and $\Delta C(t)$ represent the fitness function of the processing cost, quality loss cost, and sensitivity cost of the tolerance loop, respectively, and $\Delta\tau_{ij}$ is the torsor parameters on the j-th functional surface of the i-th part in the tolerance loop.

The mathematical model of deviation optimal allocation is established as follows:

$$\begin{aligned} \min : \ & C(t) = \sum_i \sum_j [\lambda_1 \cdot C_t(\Delta\tau_{ij}) + \lambda_2 \cdot L_t(\Delta\tau_{ij}) + \lambda_3 \cdot \Delta C(\Delta\tau_{ij})] \\ \text{s.t.} : \ & t_{imin} \le f_j(\Delta\tau_{ij}) \le t_{imax}, \ f(\Delta\tau_{ij}) \le GFR, \ \{\Omega_k, D_k\}^T = G \end{aligned} \tag{52}$$

where $\lambda_1$, $\lambda_2$, and $\lambda_3$ are the influence coefficients of processing cost, quality loss cost, and sensitivity cost, respectively, on the deviation value of the component links, and their sum is 1; $f_j(\Delta\tau_{ij})$ is the processing capability deviation of the SDT deviation of each functional surface; and $f(\Delta\tau_{ij})$ is the geometric function requirement.

A modified genetic bat algorithm [28] is used to solve the multi-objective optimization problem of the SDT parameters, as shown in Figure 16. One of the main advantages of the Bat Algorithm (BA) is that it can achieve rapid convergence of the algorithm. However, it will also lead to premature stagnation. As a result, it is easy to fall into a local optimum like many other heuristic algorithms. In the genetic operator of a genetic algorithm (GA), the selection operator improves global convergence and computational efficiency, the crossover operator transmits the optimal solution to the next generation and generates new individuals with more complex structures, and the mutation operator is an important way to ensure the special diversity of GA. GA has good diversity and searchability. The population crossover and mutation mechanism of GA is introduced into BA to form a modified genetic bat algorithm (GBA) to ensure the enhancement of diversity in the population iteration process and improve the global search efficiency and convergence of the algorithm.

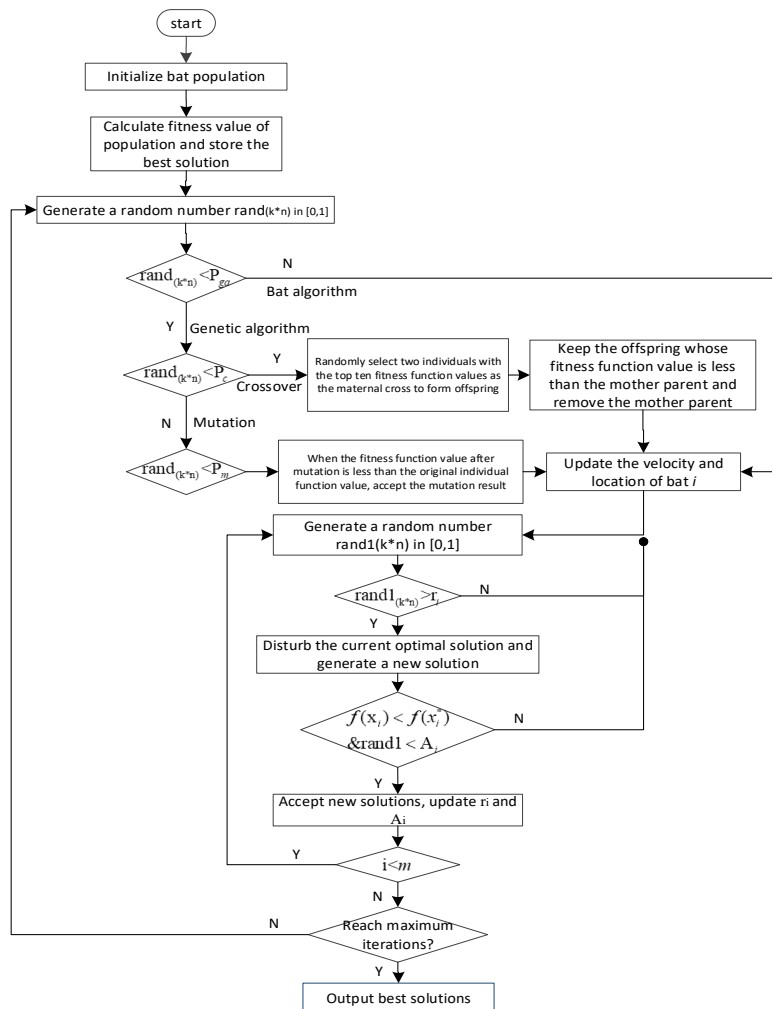

**Figure 16.** Flowchart of solving the 3D tolerance mathematical model with a genetic bat algorithm.

The modeling of three-dimensional tolerance is to map the variational geometric constraints of the tolerance zone to the deviation domain, and the solution by an intelligent algorithm is to map the optimization result of the deviation domain to the tolerance zone, and finally determine the optimal tolerance allocation value of each component link.

## 5. Case Study

An assembly composed of three rectangular parts is used to verify the effectiveness of the three-dimensional tolerance modeling method proposed in this paper. For the convenience of calculation, the components of the assembly are named as part 1, part 2, and part 3 whose functional surfaces are all planes and the nominal sizes of each are known, as shown in Figure 17.

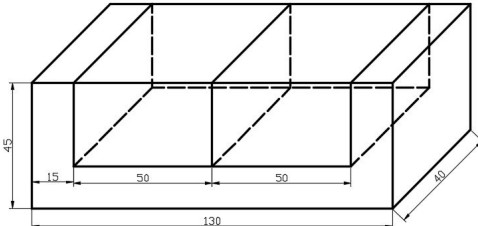

**Figure 17.** Schematic diagram of assembly nominal size.

### 5.1. Set up a Coordinate System

First, the cartesian coordinate system needs to be set up. The method of establishing the coordinate system has been introduced in Section 2. The global coordinate system is established in the assembly layer, and the local coordinate system is established in each functional surface. According to MGDE, the minimum geometric elements of the functional feature surface can be determined. The functional surface of the part is a plane. From Table 3, the MGDE of the plane is a plane, and because the shape of the plane is symmetric, the origin of the coordinates was set is at the center of the intersection of the diagonals of the rectangular planes, and the positive z axis points to the outside of the part body where the functional surface is located. The established coordinate system is shown in Figure 18.

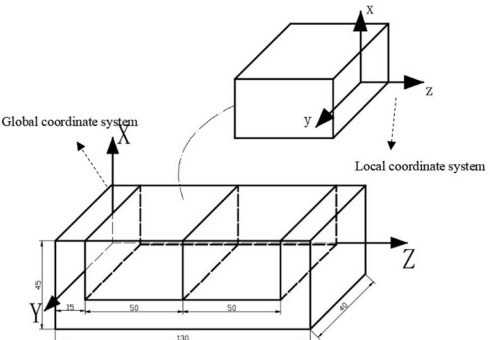

**Figure 18.** The establishment of global coordinate system and local coordinate system.

Analyzing the assembly, there are nine functional surfaces with variational geometric constraint of the three parts, and they are all plane invariance degrees, with five contact clearance domains between each functional surface. The functional feature surface of the assembly is labeled and the corresponding assembly network is established.

### 5.2. Generation of Assembly Tolerance Loop

According to Figure 19, the assembly network can be established as shown in Figure 20, following Figure 15. In Figure 20, the solid line represents the contact or clearance fit between the two functional surfaces, and there is a geometric deviation constraint. The double arrow dashed line is the geometric functional requirement proposed in this study case. In this case, it will be analyzed in the case ensuring the assembly clearance between the surface 3 of the part 1 and the surface 3 of the part 3 is less than 0.01. It can be seen that the construction of the assembly network has a significant effect on the generation

of tolerance loops. According to Figure 20, the tolerance loop expression based on the constraint relationship of functional surface is written as follows:

$$L = \left\{ [(1,1), g_1], [(1,2,3), g_2], [(1,3,3), g_3], (3,1,1) \right\} \tag{53}$$

where (1,1) represents the deviation produced by functional surface 1 on part 1, (1,2,3) represents the deviation produced on the left functional surface 1 and right functional surface 3 of part 2, $g_1$ is the deviation clearance between the functional surface 1 of the part 1 and the functional surface 1 of the part 2, and so on.

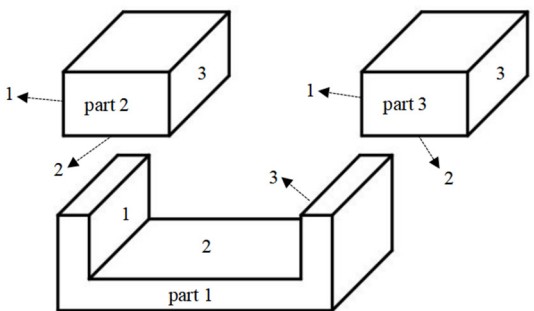

**Figure 19.** Marks on each functional surface.

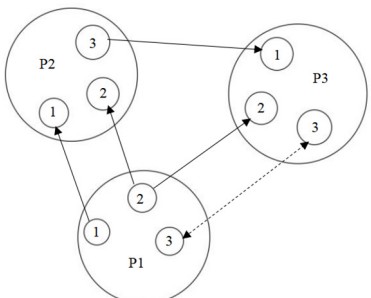

**Figure 20.** Assembly network diagram.

If two functional surfaces are located on the same part, because they already have constraints, the geometric deviation is not considered and it is recorded as 0.

Equation (53) is expressed in the form of SDT parameters:

$$\Delta\tau_{1,1} \to \Delta\tau_{1,1/2,1} \to \Delta\tau_{2,1} \to \Delta\tau_{2,3} \to \Delta\tau_{2,3/3,1} \to \Delta\tau_{3,1} \to \Delta\tau_{3,3} \to \Delta\tau_{3,3/1,3} \to \Delta\tau_{1,3} \to \Delta\tau_{1,3/1,1} = 0 \tag{54}$$

where $\Delta\tau_{2,1}$ is the SDT parameter expression of the deviation produced by the functional surface 1 of part 2, $\Delta\tau_{2,3/3,1}$ is the contact or fit clearance between surface 3 of part 2 and surface 1 of part 3, that is, the SDT parameter expression of $g_2$ in Equation (53), and so on.

Table 1 shows the torsor expressions corresponding to various tolerance zones and their variational geometric constraint inequalities. The degrees of freedom of the plane type are the rotation along the x and y axes and translation along the z axis, so each deviation can be expressed as the following equation:

$$\Delta\tau_{1,1} = \left\{ \Delta\alpha_{1,1}, \Delta\beta_{1,1}, 0, 0, 0, \Delta w_{1,1} \right\} \tag{55}$$

The geometric function requirements are the closed loop requirements in the two-dimensional dimensional chain, and the corresponding dimensional chain can be drawn according to the tolerance loop, as shown in Figure 21. The positive and negative of the deviation domain of each component link is determined according to the principle of the arrow method. Starting from one of the two functional surfaces of functional requirements, go around the dimensional chain once, and the same direction is positive. Equation (54) can be written as the following equation:

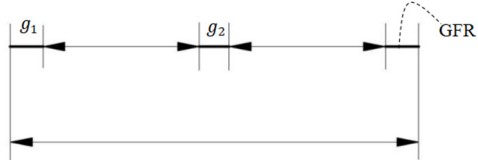

**Figure 21.** Deviation dimension chain.

$$\Delta\tau_{1,1} + \Delta\tau_{1,1/2,1} - \Delta\tau_{2,1} + \Delta\tau_{2,3} + \Delta\tau_{2,3/3,1} - \Delta\tau_{3,1} + \Delta\tau_{3,3} + \Delta\tau_{3,3/1,3} - \Delta\tau_{1,3} = 0 \quad (56)$$

Substituting Equation (56) into Equation (9), we can obtain the deviation accumulation expression of SDT deviation parameters in the global coordinate system:

$$\Delta\alpha_{1,1} + \Delta\alpha_{1,1/2,1} - \Delta\alpha_{2,1} + \Delta\alpha_{2,3} + \Delta\alpha_{2,3/1,3} - \Delta\alpha_{3,1} + \Delta\alpha_{3,3} + \Delta\alpha_{3,3/1,3} - \Delta\alpha_{1,3} = 0$$
$$\Delta\beta_{1,1} + \Delta\beta_{1,1/2,1} - \Delta\beta_{2,1} + \Delta\alpha_{2,3} + \Delta\beta_{2,3/1,3} - \Delta\beta_{3,1} + \Delta\beta_{3,3} + \Delta\beta_{3,3/1,3} - \Delta\beta_{1,3} = 0$$
$$\begin{aligned}
\Delta w_{1,1} + \Delta w_{1,1/2,1} - &\left[\Delta w_{2,1} + \Delta\alpha_{2,1}\cdot(Y_{2,1} - Y_{1,1}) - \Delta\beta_{2,1}\cdot(X_{2,1} - X_{1,1})\right]\\
+ &\left[\Delta w_{2,3} + \Delta\alpha_{2,3}\cdot(Y_{2,3} - Y_{1,1}) - \Delta\beta_{2,3}\cdot(X_{2,3} - X_{1,1})\right]\\
+ &\left[\Delta w_{2,3/3,1} + \Delta\alpha_{2,3/3,1}\cdot(Y_{2,3/3,1} - Y_{1,1}) - \Delta\beta_{2,3/3,1}\cdot(X_{2,3/3,1} - X_{1,1})\right]\\
- &\left[\Delta w_{3,1} + \Delta\alpha_{3,1}\cdot(Y_{3,1} - Y_{1,1}) - \Delta\beta_{3,1}\cdot(X_{3,1} - X_{1,1})\right]\\
+ &\left[\Delta w_{3,3} + \Delta\alpha_{3,3}\cdot(Y_{3,3} - Y_{1,1}) - \Delta\beta_{3,3}\cdot(X_{3,3} - X_{1,1})\right]\\
+ &\left[\Delta w_{3,3/1,3} + \Delta\alpha_{3,3/1,3}\cdot(Y_{3,3/1,3} - Y_{1,1}) - \Delta\beta_{3,3/1,3}\cdot(X_{3,3/1,3} - X_{1,1})\right]\\
- &\left[\Delta w_{1,3} + \Delta\alpha_{1,3}\cdot(Y_{1,3} - Y_{1,1}) - \Delta\beta_{1,3}\cdot(X_{1,3} - X_{1,1})\right] = 0
\end{aligned}$$

$$(57)$$

*5.3. Establish a Mathematical Model of Deviation Optimization*

5.3.1. Objective Function

All functional surfaces are planes. The cost function of tolerance in this paper is the cost function of plane feature dimension tolerance [28] with 45# steel as materials:

$$C(T) = 5.0261e^{-15.8903T} + \frac{T}{0.3927T + 0.1176} \quad (58)$$

Use the SDT deviation parameters to express the above equation as a processing cost–deviation function:

$$\begin{aligned}
C(T) = C(f(\Delta\tau)) = &\left(5.0261e^{-15.8903\Delta\alpha} + \frac{\Delta\alpha}{0.3927\Delta\alpha + 0.1176}\right)\\
\cdot &\left(5.0261e^{-15.8903\Delta\beta} + \frac{\Delta\beta}{0.3927\Delta\beta + 0.1176}\right) \cdot \left(5.0261e^{-15.8903\Delta w} + \frac{\Delta w}{0.3927\Delta w + 0.1176}\right)
\end{aligned}$$

$$(59)$$

The quality loss cost function (refer to Equation (31)) is a quadratic equation with a constant coefficient. This coefficient refers to the degree of influence of different tolerances on the cost of quality loss, so it can be regarded as a constant. This paper used a constant value of 60 [23]. Therefore, the quality loss function is:

$$L(T) = 60T^2 \quad (60)$$

Using the SDT deviation parameters to express the above equation as a quality loss cost–deviation function:

$$L(T) = L(f(\Delta\tau)) = 60\cdot\left(\Delta\alpha^2 + \Delta\beta^2 + \Delta\gamma^2\right) \quad (61)$$

The sensitivity–deviation function is established on the basis of the processing cost function, so the sensitivity–deviation function expressed by the SDT deviation parameters can be written as:

$$\Delta C(T) = \Delta C(f(\Delta\tau)) = \left[ -79.8662e^{-15.8903\Delta\alpha} + \frac{0.1176}{(0.3927\Delta\alpha + 0.1176)^2} \right] \cdot$$
$$\left[ -79.8662e^{-15.8903\Delta\beta} + \frac{0.1176}{(0.3927\Delta\beta + 0.1176)^2} \right] \cdot \left[ -79.8662e^{-15.8903\Delta\alpha} + \frac{0.1176}{(0.3927\Delta\beta + 0.1176)^2} \right] \tag{62}$$

### 5.3.2. Constraints

The component links of the assembly dimensional chain all have a certain range of processing capabilities. To make it economical, that is, each deviation component is given a constraint range, which is usually formulated on the basis of some experience. The parts in this paper are relatively simple; according to the processing capacity of the machine tool and other factors, the boundary condition range of each SDT component is taken as:

$$-0.4 \leq \Delta\tau_{ij} \leq 0.4 \tag{63}$$

The geometric functional requirement constraint is to ensure that the assembly clearance between the functional surface 3 of part 1 and the functional surface 3 of part 3 is less than 0.01, that is:

$$\Delta\tau_{3,3/1,3} \leq 0.01 \tag{64}$$

Assembly tolerance constraints refer to Equations (56) and (57).

### 5.3.3. Mathematical Model of Optimal Deviation Allocation

The mathematical model of this example in this paper is:

$$\min : \sum_i \sum_J \lambda_1 \cdot C(f(\Delta\tau_{ij})) + \lambda_2 \cdot L(f(\Delta\tau_{ij})) + \lambda_3 \cdot \Delta C(f(\Delta\tau_{ij}))$$
$$\text{s.t} : -0.4 \leq \Delta\tau_{ij} \leq 0.4$$
$$\Delta\tau_{3,3/1,3} \leq 0.01 \tag{65}$$
$$\Delta\tau_{1,1} + \Delta\tau_{1,1/2,1} - \Delta\tau_{2,1} + \Delta\tau_{2,3} + \Delta\tau_{2,3/3,1} - \Delta\tau_{3,1} + \Delta\tau_{3,3} + \Delta\tau_{3,3/1,3} - \Delta\tau_{1,3} = 0$$

### 5.4. Solve the Model with Genetic Bat Algorithm

In this paper we improved the genetic bat algorithm in MATLAB to solve the mathematical model of multi-objective optimization [29]. The population sum of the three deviation directions is set to 50, and the maximum iterations to 200. Then, we can obtain the deviation variation values of the nine functional feature surfaces in the three directions of $\Delta\alpha$, $\Delta\beta$, and $\Delta\gamma$, as shown in Table 4.

**Table 4.** Deviation values of each plane feature.

| Type | p1,1 | p1,1/2,1 | p2,1 | p2,3 | p2,3/3,1 | p3,1 | p3,3 | p3,3/1,3 | p1,3 |
|------|------|----------|------|------|----------|------|------|----------|------|
| $\Delta\alpha1$ | −0.0626 | 0.0686 | 0.1339 | 0.0666 | 0.0750 | −0.0711 | 0.2132 | −0.2062 | 0.3365 |
| $\Delta\alpha2$ | −0.1070 | −0.1661 | 0.0687 | 0.0328 | 0.0769 | 0.3695 | 0.0554 | −0.0226 | 0.0033 |
| $\Delta\alpha3$ | −0.0083 | 0.0885 | −0.0164 | 0.1955 | −0.0300 | 0.2892 | −0.2052 | −0.2146 | 0.0073 |
| $\Delta\alpha4$ | −0.0598 | −0.0036 | 0.2237 | 0.1418 | 0.3842 | 0.0350 | 0.2310 | −0.1199 | 0.3371 |
| $\Delta\beta1$ | 0.2322 | −0.0184 | 0.2598 | −0.2113 | 0.3229 | −0.0298 | −0.0993 | −0.0328 | −0.0641 |
| $\Delta\beta2$ | −0.1472 | 0.3395 | −0.1493 | −0.1989 | 0.2057 | −0.1731 | −0.0340 | −0.1542 | 0.1300 |
| $\Delta\beta3$ | −0.1926 | 0.0816 | 0.1306 | 0.2561 | −0.0248 | 0.1936 | 0.3226 | −0.0024 | 0.3058 |
| $\Delta\beta4$ | −0.0019 | 0.3776 | −0.1030 | 0.1998 | 0.2418 | 0.13816 | 0.1633 | −0.1097 | 0.1772 |
| $\Delta w1$ | 0.0558 | −0.1005 | 0.0327 | 0.1723 | 0.1229 | 0.1767 | −0.1343 | −0.1327 | 0.0079 |
| $\Delta w2$ | 0.3817 | 0.0896 | 0.1699 | −0.1589 | 0.3161 | −0.1238 | −0.1077 | −0.1144 | 0.3893 |
| $\Delta w3$ | 0.0312 | 0.0115 | −0.1237 | 0.1247 | 0.3588 | −0.2559 | 0.1361 | −0.0632 | 0.0810 |
| $\Delta w4$ | −0.0037 | −0.0810 | 0.2691 | 0.3744 | −0.1031 | 0.1071 | −0.0366 | −0.0150 | −0.0700 |

The value after each deviation in the table represents the variation of the four vertices of the same plane in the direction. We know that the data in the table are the values of the deviation domain. For example, $\Delta\alpha1$ is the deviation caused by vertex 1 rotating around

the x axis. In order to obtain the tolerance value, the deviation values need to be mapped to the tolerance values. It can be seen from the dimensional chain in Figure 21 that only the values in z direction affects the assembly clearance, so the tolerance values in the z-axis direction are studied separately.

From Table 5, the upper and lower deviation values of each assembly dimension chain can be obtained. The length of functional surface 1 of part 1 is $100^{+0.771}_{-0.0737}$, the length of functional surface 2 of part 2 is $50^{+0.6435}_{-0.2826}$, and the length of the functional surface 2 of the part 3 is $50^{+0.3128}_{-0.3902}$. At the same time, the dimensional tolerance values of other surfaces on the part can also be obtained, which will not be discussed in detail here. If you need to calculate the cost value of the objective function, you only need to substitute the deviation values into the mathematical model to calculate. The above case verifies the feasibility and effectiveness of the optimal 3D tolerance allocation in this paper. By mapping the tolerance domain and the deviation domain to each other, we can obtain the influence of the 3D tolerance chain on the 2D dimension chain and more practical design tolerances.

**Table 5.** Deviation values of each plane feature.

| Functional Surface | $T_{min}$ | $T_{max}$ |
| :---: | :---: | :---: |
| p1,1 | −0.0037 | 0.3817 |
| p1,1/2,1 | −0.1005 | 0.0896 |
| p2,1 | −0.1237 | 0.2691 |
| p2,3 | −0.1589 | 0.3744 |
| p2,3/3,1 | −0.1031 | 0.3588 |
| p3,1 | −0.2559 | 0.1767 |
| p3,3 | −0.1343 | 0.1361 |
| p3,3/1,3 | −0.1327 | −0.0150 |
| p1,3 | −0.0700 | 0.3893 |

In order to verify the validity and feasibility of the multi-objective function used in this paper, this algorithm was also used to solve the problem with a single objective function to obtain the tolerance result. The results of mapping the deviation domain of the single objective function to the tolerance domain are shown in Table 6.

**Table 6.** Result of the single objective function.

| Functional Surface | $T_{min}$ | $T_{max}$ |
| :---: | :---: | :---: |
| p1,1 | −0.3724 | 0.3378 |
| p1,1/2,1 | −0.3043 | 0.2805 |
| p2,1 | −0.3889 | 0.2604 |
| p2,3 | −0.2918 | 0.1297 |
| p2,3/3,1 | −0.2882 | 0.2972 |
| p3,1 | −0.2517 | 0.3030 |
| p3,3 | −0.1318 | 0.3349 |
| p3,3/1,3 | −0.3451 | −0.1162 |
| p1,3 | −0.3870 | 0.2681 |

The tolerance, $C(T)$, $L(T)$, $\Delta C(T)$, and the total cost are shown in Table 7.

The cost of the multi objective function was less than that of the single objective function. In real processes, the calculation results of the single objective function do not consider the influence of quality and sensitivity on the parts. If a batch of products does not consider enough economic factors in the optimization design, nearly 20% of the parts will be damaged before the expected loss period, which further proves the importance of quality loss in production compared to the total cost of products.

**Table 7.** Total cost of different objective functions.

|  | | p1,1 | p1,1/2,1 | C(T) | L(T) | ΔC(T) | Total Cost |
|---|---|---|---|---|---|---|---|
| Result of the single objective function | $T_{1,2}$ | 1.3653 | 1.8829 | 111.8426 | 56.6813 | |
| | $T_{2,2}$ | 1.0708 | 2.6002 | 68.7968 | 92.6071 | 493.6342 |
| | $T_{3,2}$ | 1.0214 | 2.7176 | 62.5955 | 93.9102 | |
| Result of the multi objective function | $T_{1,2}$ | 1.0096 | 2.7503 | 61.1575 | 94.5597 | |
| | $T_{2,2}$ | 1.2587 | 2.0571 | 95.0595 | 58.1012 | 477.9916 |
| | $T_{3,2}$ | 0.7582 | 3.2887 | 34.4920 | 126.5256 | |

## 6. Conclusions

Tolerance design is a key problem in manufacturing processes. An optimization of three-dimensional tolerance allocation in the conceptual design stage was proposed in this paper to make the tolerance design more adaptive to 3D models of real parts and processed. Combined with HTM and SDT, the optimization problem of three-dimensional tolerances can be transformed into a optimization problem of torsor allocation in the deviation space to solve the 3D tolerance allocations. The numerical simulation and results expounding the complete process of the method are shown in this paper.

Future studies are planned to integrate the deformation of parts caused by environmental factors in the actual working process and analyze the influence of actual working conditions on the manufacturing of parts.

**Author Contributions:** Conceptualization, J.Y.; methodology, Y.C.; software, K.Y.; validation, K.Y. and Y.G.; formal analysis, Y.G.; investigation, Z.W.; resources, Z.W.; data curation, K.Y.; writing—original draft preparation, K.Y.; writing—review and editing, Z.W.; visualization, K.Y.; supervision, Z.W.; project administration, Z.W.; funding acquisition, Z.W. All authors have read and agreed to the published version of the manuscript.

**Funding:** This research was funded by the National Key R&D Program of China (Grant No. 2022YFB3304203 and Grant No. 2019YFB1707101) and National Natural Science Foundation of China (Grant No. 52175520) . This research was also supported by the Anhui Key Laboratory of Mine Intelligent Equipment and Technology (Grant No. KSZN202001001).

**Data Availability Statement:** The data presented in this study are available on request from the corresponding author.

**Conflicts of Interest:** The authors declare no conflict of interest.

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
