# Peer review of "Optimization of 3D Tolerance Design Based on Cost–Quality–Sensitivity Analysis to the Deviation Domain"

_2673-4052, doi:10.3390/automation4020009_

Round 1

Reviewer 1 Report

The paper on 3D Tolerance design based on a modified bat algorithm, by K.Yang et al., presents a sound analysis of GPS optimization applying a search algorithm. It is  based on mapping  the geometric variations of the tolerance zone to the deviation domain, and analyzing the cost sensitivity to the deviation domain. The complex method is detailed, and a study case is shown.

The paper topic is very interesting as an analysis method and due to its clear application in helping to make complex tolerance costs analysis. The paper is well presented and the discussion is sound with important information, details and figures to support the presentation. No doubt this paper deserves publication. However, there are flaws and details to consider seriously. If the Authors make a sound revision based on the comments, the paper would contain enough and important missing information and the new version could be accepted.  I do encourage the Authors to read the next comments positively, focusing on improving the paper. 

Note, please, that the revision of all the mathematical expressions is limited, and it is up to the Authors to double-check the details to ensure the best quality and avoid typos or mistakes. It seems mostly correct. Indeed, typo details would not affect the presented results.

The language has many flaws in the whole text. It should be revised thoroughly.

The review comments (14 pages) are attached.

Reviewer 2 Report

The authors present a 3D Tolerance design based on a modified bat algorithm.

The paper is well organized and easy to follow. I have few comments:

1. The abstract and conclusion must be improved showing what is done and what is achieved from the research.

2. The literature review must be improved by showing more pieces of research and presenting a deep discussion.

3. Based on the second point, the novelty of the work can be clearly presented.

4. Please write the paper according to the journal format.

5. The authors can mention some details about future research.

Round 2

Reviewer 1 Report

The Authors have made a thorough revision, considering positively the comments and points raised during the revision. Unfortunately, some points are almost the same as before, not adding the required extra information that could really make a difference, for instance, on the points below. Please, consider completing the draft with the missing information to get a much improved draft.

-- eq-16 & 36: omitting the quadratic term...
if it is the case, comment clearly how/why, kind of: considering as adequate the linear approximation to the quadratic term...

--Figure 13: making any brief comment about what is found in the figure.

--L804: a reference to the modified genetic bat algorithm method used.

Note also that in the abstract there is a mention of an 'improved' algorithm. Therefore, this has to be explained with some comments about what these improvements are.  

-- Section 5.3.1: make a direct mention to the ref-28 since the used cost function must be justified, kind of:
(L909) The cost function of tolerance in this paper is the function of plane feature dimension tolerance found in [28] with 45# steel as materials and included in Table-2: (check it is one case mentioned in table-2 !? )

eq-59: what equations in the paper provide eq-59 from eq-58 ? The same for eq-62.

-- section 5.4
relating clearly the information on tables 4 and 5 to the examples provided in lines 991-ff: even if obvious to the Authors, the numerical cases must be fully justified at least in one case (length of functional surface 1 of part 1 is... ), and left to the Readers other solutions (... length of functional surface 2 of part 2 is ... the length of the functional surface 2 of the part 3 is...)    

Also, include in the paper the clear and key messages provided in the answers by the Authors:
T_min is the lower limit of tolerance and T_max is the upper limit of tolerance. If the tolerance is smaller than T_min, it would cause more cost for the manufacturer. If the tolerance is bigger than T_max, the part could not meet the function requirement.
